# Lung Epithelial Cells from Obese Patients Have Impaired Control of SARS-CoV-2 Infection

**DOI:** 10.3390/ijms24076729

**Published:** 2023-04-04

**Authors:** Mellissa Gaudet, Eva Kaufmann, Nour Jalaleddine, Andrea Mogas, Mahmood Hachim, Abiola Senok, Maziar Divangahi, Qutayba Hamid, Saba Al Heialy

**Affiliations:** 1Meakins-Christie Laboratories, Research Institute of the McGill University Healthy Center, Montreal, QC H4A 3J1, Canada; 2Department of Biomedical and Molecular Sciences, Queen’s University, Kingston, ON K7L 3N6, Canada; 3College of Medicine, Mohammed Bin Rashid University of Medicine and Health Sciences, Dubai P.O. Box 505055, United Arab Emirates; 4Sharjah Institute for Medical Research, University of Sharjah, Sharjah P.O. Box 27272, United Arab Emirates

**Keywords:** SARS-CoV-2, infection, obesity, immune response, ACE2, transcriptome profiling, bronchial epithelial cells

## Abstract

Obesity is known to increase the complications of the COVID-19 coronavirus disease caused by severe acute respiratory syndrome coronavirus 2 (SARS-CoV-2). However, the exact mechanisms of SARS-CoV-2 infection in obese patients have not been clearly elucidated. This study aims to better understand the effect of obesity on the course of SARS-CoV-2 infection and identify candidate molecular pathways involved in the progression of the disease, using an in vitro live infection model and RNA sequencing. Results from this study revealed the enhancement of viral load and replication in bronchial epithelial cells (NHBE) from obese subjects at 24 h of infection (MOI = 0.5) as compared to non-obese subjects. Transcriptomic profiling via RNA-Seq highlighted the enrichment of lipid metabolism-related pathways along with *LPIN2*, an inflammasome regulator, as a unique differentially expressed gene (DEG) in infected bronchial epithelial cells from obese subjects. Such findings correlated with altered cytokine and angiotensin-converting enzyme-2 (ACE2) expression during infection of bronchial cells. These findings provide a novel insight on the molecular interplay between obesity and SARS-CoV-2 infection. In conclusion, this study demonstrates the increased SARS-CoV-2 infection of bronchial epithelial cells from obese subjects and highlights the impaired immunity which may explain the increased severity among obese COVID-19 patients.

## 1. Introduction

Infection by severe acute respiratory syndrome coronavirus 2 (SARS-CoV-2) causes a wide range of clinical symptoms of variable degrees. The severity of COVID-19 has been shown to be associated with age, gender and the presence of comorbidities such as diabetes, hypertension and obesity. Numerous studies have shown that obesity increases the risk for hospitalization and mortality rate in COVID-19 patients [1]. This is particularly alarming because rates of obesity are high worldwide, with 40% of adults being obese in the United States (US) according to the Centers for Disease Control and Prevention (CDC) [2,3], making obesity one of the most important risk factors in COVID-19. In a retrospective study in France, out of 124 COVID-19 patients admitted to the intensive care unit (ICU), 75.8% were obese [4]. The same trends have been observed in Wuhan, China, where 68% of COVID-19 patients were overweight/obese and 37% of these patients showed severe symptoms [5]. Extensive investigations have been conducted to understand this association between obesity and severe COVID-19, with the goal of ameliorating COVID-19 treatment in obese patients.

It is well documented that SARS-CoV-2 binds to angiotensin-converting enzyme-2 (ACE2), expressed in a wide variety of human structural cells in kidney, lung, thyroid and adipose tissue [6]. Despite its role in viral entry, ACE2 plays a protective role in the cardiovascular system. Physiologically, ACE2 is a counter-regulator of the renin-angiotensin system (RAAS) and is involved in regulating systemic blood pressure. We have previously shown that bronchial epithelial cells of obese patients express higher levels of *ACE2*, which may explain the increased susceptibility to infection and severe outcomes in obese patients [7]. We further highlighted how obesity-associated *ACE2* polymorphisms may enhance COVID-19 severity in obese patients [8]. Of note, obesity is associated with chronic systemic inflammation. This is important, as inflammation is associated with severity of COVID-19 [9]. This was shown in one study, where a high rate of hospitalization among obese COVID-19 patients can be explained by their low-grade inflammatory response [10]. Other studies showed that obese COVID-19 patients have higher levels of inflammatory mediators, such as D-dimer and C-reactive protein, which also contributed to the severity of COVID-19 [11].

Despite extensive research that highlighted the strong association between obesity and COVID-19 disease severity, the underlying mechanisms that link obesity to SARS-CoV-2 infection are yet to be defined. To our knowledge, this is the first study that demonstrates the involvement of obesity as a major contributor to enhanced SARS-CoV-2 load in a time-dependent manner using an in vitro live infection model of normal human bronchial epithelial (NHBE) cells. Our in vitro model displayed higher viral entry and replication in NHBE cells from obese subjects as compared to non-obese subjects—specifically, at 4 h and 24 h post-infection. The difference in viral propagation was further explained by transcriptomic analysis that revealed *LIPN2*, an inflammasome regulator, as a unique gene to NHBE cells from obese subjects. These findings correlated with altered cytokine expression that exhibited downregulated levels at early stages of the infection, following a decrease (24 h) and then an increase (72 h), in NHBE cells from obese subjects. This study provides a molecular overview on the association of obesity and increased disease severity in obese COVID-19 patients via the modulation of the immune response.

## 2. Results

### 2.1. Increased Viral Detection and Replicates in Bronchial Epithelial Cells from Obese Subjects

Bronchial epithelial cells (NHBE) from non-obese and obese subjects were infected with SARS-CoV-2 at an MOI of 0.5 for a series of time points. The internalized virus was detected by measuring the viral nucleocapsid (N) RNA via qRT-PCR. Viral replication was also determined by qRT-PCR by measuring the mRNA levels of viral envelope transcripts. Both viral internalization and replication were comparable between NHBE cells of non-obese and obese subjects before 24 h of infection (Figure 1A,B). An oscillation of these 2 viral genes were observed throughout time, where an initial increase in mRNA expression was seen at 4 h and 8 h, followed by a decrease at 12 h and, once again, an increase at 18 h. Interestingly, a significant increase in viral detection and replication was observed at the 24 h time point in cells from obese subjects compared to non-obese subjects. However, an important drop in the 2 viral markers nucleocapsid (N) and viral envelope (UpE) was detected at 48 h and 72 h in both subject groups.

### 2.2. Enrichment of Lipid Metabolism-Related Pathways in Infected Bronchial Epithelial Cells from Obese Subjects

Since our data displayed an increase in viral entry and replication at 4 h, with a significant difference between obese and non-obese subject groups at 24 h, we aimed to investigate the transcriptomic signature of the bronchial epithelial (NHBE) cells from non-obese and obese subjects at 4 h and 24 h, prior and post-SARS-CoV-2 infection. This will help explain the difference seen in the viral load among non-obese and obese subjects. Firstly, we generated a baseline transcriptome profile via individualized differentially expressed genes (iDEGs) to identify the DEGs between non-infected NHBE cells from obese and non-obese (Figure 2). For each time point, DEGs between obese and non-obese were identified and intersected with DEGs identified in the other time point. 1234 DEGs were identified between obese and non-obese NHBE cells at 4 h, whereas 1644 DEGs were identified at the 24 h time point. Interestingly, 286 genes were differentially expressed in obese versus non-obese subjects in both time points, indicating core transcriptome difference between the 2 patient groups (Figure 2A). These 286 genes were mainly enriched in pathways related to lipid metabolism (glycerolipid biosynthetic process, triglyceride biosynthesis, positive regulation of lipid metabolic process, phosphatidylglycerol metabolic process and protein lipidation), cell shape and structure (regulation of cell shape, apical junction assembly and cilium assembly) and in pathways involved in infection and tissue repair (epithelial to mesenchymal transition in colorectal cancer, negative regulation of wound healing, herpes simplex virus 1 infection and negative regulation of response to external stimulus), as shown in Figure 2B.

Secondly, we furthered our analysis to identify the DEGs between infected and non-infected NHBE cells from obese and non-obese subjects. The DEGs at each time point, 4 h and 24 h, were intersected to identify the core transcriptome differentially expressed genes. Our results displayed 41 DEGs between non-infected and infected, and in obese and non-obese subject groups (Figure 3A). Interestingly, these 41 DEGs were enriched in pathways related to metabolic, cellular and developmental processes (Figure 3B). Highlighting the significant increase of viral replication and load that was only seen in NHBE from obese subjects, our data presented 728 DEGs that were unique to infected obese patients’ bronchial epithelial cells versus those of non-infected patients at 24 h. Interestingly, those DEGs were enriched in pathways such as those regulating the metabolism of lipid and type I interferon, among other significantly enriched pathways (Figure 3A,C).

### 2.3. LPIN2 as a Unique DEG in Infected Bronchial Epithelial Cells from Obese Subjects

To identify key genes that are unique to bronchial epithelial cells from obese subjects and are core players in SARS-CoV-2-infected cells as compared to non-infected cells, we intersected the 2 initial comparisons which comprise the 286 DEGs (unique to obesity) and the 41 DEGs (unique to SARS-CoV-2 infection). Only five genes (*CASC5 (KNL1)*, *LPIN2*, *MIR143HG (CARMN)*, *ULBP1* and *THRB*) were identified (Figure 4). Interestingly, among the five genes identified, *LPIN2*, a magnesium-dependent phosphatide phosphatase enzyme, is known to play a major role in triglyceride metabolism, inflammasome regulation, and as a transcriptional co-regulator of gene expression [12,13,14,15].

To gain insight about the expression of *LPIN2* in bronchial epithelial cells during the infection, the normalized gene expression of *LPIN2* was extracted from the RNA-Seq data at different time points, in both non-obese and obese subjects. The percentage of change of *LPIN2* expression in infected compared to non-infected NHBE cells was plotted as a percentage against the time points in both non-obese and obese groups (Figure 5). At 4 h of SARS-CoV-2 infection, the NHBE cells from non-obese subjects exhibited higher *LPIN2* expression as compared to the obese subject group (Figure 5). However, at 24 h post-infection, *LPIN2* expression was decreased in the non-obese subject group and was rather increased in obese subject groups (Figure 5). Interestingly, *LPIN2* expression was retained in the infected NHBE cells from non-obese subjects after 24 h, to reach a maximal expression peak at 48 h post-infection (Figure 5), while those of obese subject group failed to overexpress *LPIN2* after this timepoint. Such a trend of differential expression of *LPIN2* might coincide with the increased viral entry and replication in NHBE cells. Additionally, this may indicate a possible modulation of the inflammatory response upon SARS-CoV-2 infection given the role of *LPIN2* in the regulation of inflammation.

### 2.4. Bronchial Epithelial Cells from Obese Subjects Mount a Stronger Cytokine Response upon SARS-CoV-2 Infection at Early Stages of Infection

Having pointed out *LPIN2* as a unique DEG in SARS-CoV-2-infected NHBE cells, the inflammatory profile was assessed during infection by qRT-PCR in an in vitro culture system. The mRNA expression of five inflammatory markers (*IL-17A*, *IL-1β*, *IL-6*, *IL-8* and *IFN-β*) was measured in bronchial epithelial cells from obese and non-obese subjects. Like *IL-8*, *IL-17A* plays a role in recruiting neutrophils to the lung [16] and in contributing to T cell activation. Our data showed that upon SARS-CoV-2 infection, NHBE cells from obese subjects displayed significantly increased *IL-17A* mRNA levels compared to all other groups at 4 h (Figure 6A). Then, *IL-17A* expression quickly decreased significantly to match the baseline levels of the cytokine in infected and non-infected cells from non-obese subjects, and non-infected cells from obese individuals, at the 8 h time point. However, this expression of *IL-17A* mRNA was increased at 72 h in both infected non-obese and obese subject groups as compared to other time points (Figure 6A).

Known for its role in downregulating the inflammatory response and for its antiviral properties, the mRNA expression of *IFN-β* was also assessed [17,18]. Differential expression of *IFN-β* mRNA was observed between SARS-CoV-2-infected bronchial epithelial cells from non-obese and obese subjects, where the obese group displayed significantly higher *IFN-β* mRNA expression at 4 h and 8 h post-infection. At 8 h post-infection, infected cells from obese subjects are at the peak of their *IFN-β* expression while non-infected cells from the same obese subjects already express low levels of *IFN-β*. However, this *IFN-β* expression showed reduced expression levels starting at 24 h post-infection (Figure 6B). Notably, this correlates with the increase in internalization and replication of the virus in NHBE cells from obese subjects at 24 h post-infection, as seen in Figure 1. This also coincides with the changes in *LPIN2* expression seen in cells from obese subjects post-infection (Figure 5).

Elevated levels of *IL-6*, a main pro-inflammatory mediator, have been associated with a poorer prognosis in COVID-19 patients [19,20]. Independent of infection, bronchial epithelial cells from obese subjects showed a higher *IL-6* expression at 4 h than the bronchial epithelial cells from non-obese subjects (Figure 6C). From 12–48 h timepoints, *IL-6* expression in all 4 groups—infected/non-infected bronchial epithelial cells from non-obese/obese subjects—was very low and spiked again at 72 h, particularly in the infected cells from obese subjects (Figure 6C).

A significant difference of *IL-8* expression was determined between the infected non-obese and obese groups. *IL-8* mRNA expression in NHBE cells from obese subjects (4 h post-infection) was greater than that in the infected non-obese group (Figure 6D). In cells from obese subjects, *IL-8* expression was at its highest at the beginning of infection, but then it steadily declined. Notably, the expression of *IL-8* mRNA peaked in the cells from infected non-obese subjects at 72 h, while it remained at baseline throughout the assessment period (Figure 6D). *IL-8* as a neutrophil chemoattractant might contribute to the cleanup of infected cells or exacerbating tissue damage, within different timeframes in obese (earlier) vs. non-obese (later) individuals.

In contrast to the other cytokines, expression of *IL-1β* remained similar between groups throughout the assessment period. In all groups, *IL-1β* expression increased after 48 h of infection (Figure 6E). However, at 72 h post-infection, there was a significant increase of expression of *IL-1β* in the infected non-obese group as compared to all other time points of the same infected group (Figure 6E).

### 2.5. Upregulated ACE2 Expression in Bronchial Epithelial Cells from Obese Subjects

Given the differential viral load and expression of cytokines in NHBE cells, we aimed to assess the mRNA levels of *ACE2* during SARS-CoV-2 infection in the targeted cells. NHBE cells from obese subjects showed increased *ACE2* expression at 4 h of infection compared to non-infected cells as well as infected and non-infected cells from non-obese subjects (Figure 6F). At 24 h, a notable decrease in *ACE2* expression was seen in all groups. This correlates with the changes in viral internalization/replication and cytokine changes that were described earlier. *ACE2* levels in both infected non-obese and obese subjects increase again at 72 h. However, this late increase in *ACE2* expression was more pronounced in the non-obese infected group.

Collectively, this data highlights the interplay between obesity and SARS-CoV-2 infection on the progression of the disease through the modulation of the inflammatory response.

## 3. Discussion

The association between COVID-19 severity and obesity has been well documented. Several studies have now shown higher rates of hospitalization and more critical outcomes with cytokine storm [21]. However, and to our knowledge, an explanation of the exact mechanisms through which obesity confers increased susceptibility to SARS-CoV-2 is limited. In this study, we highlighted the enhanced viral load in bronchial epithelial cells in obese subjects upon SARS-CoV-2 infection using an in vitro infection model. We performed RNA-Seq analysis to profile the core transcriptomic difference between non-obese and obese subjects upon infection to better understand the enhanced viral load among the obese subjects. Interestingly, our data emphasized the enrichment of lipid metabolism-related pathways in infected bronchial epithelial cells from obese subjects, with *LPIN2*, known for inflammasome regulation, being a unique differentially expressed gene. Such findings were further translated into an in vitro application, where we have shown the dysregulation of immune responses via the altered cytokine expression in SARS-CoV-2-infected bronchial epithelial cells of obese subjects.

In a recent study, we characterized the metabolic profiling of obese COVID-19 subjects, where we highlighted the possible involvement of N6-acetyl-l-lysine and p-Cresol as two key metabolites in the pathogenesis of obese COVID-19 subjects. We showed that the elevated levels of these two metabolites could potentially enhance viral replication, on the one hand, and increase ACE2 expression, on the other, in obese subjects [22]. Previous studies suggest that SARS-CoV-2 infection subverts the host’s altered lipid metabolism and exploits metabolic and immune alterations in obesity, which promotes viral propagation and chronic inflammation [23,24]. These findings correlate with our results which showed significantly higher SARS-CoV-2 internalization and replication in bronchial epithelial cells obtained from obese subjects at 24 h. It is unsurprising that differences in gene expression that were found between non-infected bronchial epithelial cells from non-obese and obese subjects via RNA-Seq are enriched in lipid-related pathways. It has been documented that host lipids affect SARS-CoV-2 entry, replication and propagation, hence affecting the virus life cycle [25]. However, our data has also identified the expression of *LPIN2* as a unique DEG in the infected bronchial epithelial cells of obese subjects. *LPIN2*, a phosphatidic acid phosphatase-1 (PAP-1)-encoding gene, plays important roles in controlling the metabolism of fatty acids. Additionally, *LPIN2* is known to be involved in inflammation-modulating apoptosis of polymorphonuclear cells and as a negative regulator of the NLRP3 inflammasome [12,13,14,15]. Mutations in *LPIN2* can increase inflammasome activity and interleukin (IL)-1 release in primary human and mouse macrophages [26]. Of note, PAP-1 has important roles in the mediation of the lipid metabolism, affecting the replication and envelope formation of several mRNA viruses by regulating downstream lipid production, hence affecting its life cycle [27,28].

To gain further mechanistic insight on how obesity contributes to SARS-CoV-2 progression and susceptibility, we aimed to assess the cytokine expression of the bronchial epithelial cells from obese versus non-obese subjects upon the differential development of the viral titer. It is worth noting that one of the important mechanisms that links obesity and the worsening of COVID-19 disease is the dysfunctional immunity and the induction of the cytokine storm [29,30]. Inflammatory markers *IFN-β*, *IL-8*, *IL-17A* and *IL-6* displayed higher expression in bronchial epithelial cells from obese subjects. Interferons (IFNs) are the first pivotal defense line against viruses at early stages of infection and its downregulation leads to uncontrolled viral replication and delayed viral elimination [30,31]. A peak in *IFN-β* expression was observed at 8 h post-infection in cells from obese subjects. However, as time progressed, the IFN response decreased notably, starting at 12 h. Interestingly, this correlated with the detected increase in viral internalization and replication at 24 h. Clinical observations show that obese patients fail to launch a robust innate and acquired immune responses [32]. This slow response may be the cause of more severe COVID-19 disease in obese and overweight individuals. A study has shown that a delayed IFN type I response is more detrimental, causing more lung immunopathology in mice compared to that in animals that had little to no IFN type I response during SARS-CoV-1 infections [33]. Moreover, previous studies on H1N1 influenza reported an impaired production of *IFN-α* and *IFN-β* due to leptin, a hormone produced by the adipose tissue that is known to regulate fat storage [30,31].

An important drop in cytokine expression was observed during infection. This is probably related to the virus taking over the transcriptional machinery of the cell to transcribe viral genes. This would be supported by the significant increase of viral gene expression at 24 h post-infection, where the host inflammatory response is largely downregulated. Cytokine dysregulation was also shown in other studies involving H1N1 influenza virus, driving tissue damage and pathological progression. *IL-6* and *IL-1β* expression were shown to be downregulated in the lungs of obese mice at early stages of H1N1 infection when compared to their non-obese counterparts [34,35]. Despite this decrease of inflammatory markers, at 72 h post-infection, a remarkable increase of cytokine expression (*IL-17A*, *IL-6*, *IL-1β* and *IL-8*) was detected. Such an increase coincides with the downregulated expression of viral load and *LPIN2* at 72 h, as well as downregulated *IFN-β* in infected cells. Remarkably, *IL-6* was among the significant highly detected cytokines in infected bronchial epithelial cells from obese subjects. Mechanistic studies have shown leptin and adiponectin levels to be associated with the activation of inflammatory monocytes and neutrophils through NF-κB signaling pathways which, in turn, induces the secretion of several cytokines such as *IL-6*, resulting in excessive inflammatory responses or a cytokine storm [36,37].

The dysregulated immune response can also be correlated with ACE2 mRNA expression. We have previously shown that bronchial epithelial cells from obese subjects express more ACE2 mRNA at baseline [38]; hence, we wanted to evaluate the expression of ACE2 during the course of the in vitro infection. Our data show significantly higher mRNA expression of *ACE2* in the infected bronchial epithelial cells from obese subjects compared to infected cells from non-obese subjects, as early as 4 h post-infection. This was expected, as upregulated ACE2 levels in obese individuals have been well established [7,23,39]. However, this expression was seen to decrease with time post-infection, in both bronchial epithelial cells from obese and non-obese subjects. The decrease in *ACE2* expression seen at later stages post-infection is supported by other studies which have shown that SARS-CoV-2 infection decreases ACE2 expression [7,40]. After binding to SARS-CoV-2 spike protein, ACE2 fails to catalyze protective and anti-inflammatory Angiotensin (1–7), thus shifting the renin angiotensin system (RAS) to a pro-inflammatory phenotype which, in turn, leads to respiratory distress [29]. Despite this decrease in ACE2 expression, notably, our data displayed an elevated ACE2 expression in infected bronchial epithelial cells from non-obese subjects at 72 h post-infection, but not in infected bronchial epithelial cells from obese subjects. This could be attributed to obesity-dysregulated lipid metabolism via the sterol-response element-binding proteins (SREBP), a transcription factor associated with adipogenesis [38].

In summary, this is the first study to show the time course of SARS-CoV-2 infection in obese and non-obese bronchial epithelial cells, using an in vitro infection model. The study provided a molecular overview on the involvement of obesity in increased SARS-CoV-2 infection. It also emphasized the dual role of both obesity and SARS-CoV-2 as complementary factors that contribute to the detrimental progression of COVID-19. This study proves that bronchial epithelial cells from obese subjects are, in fact, infected at a higher rate than bronchial epithelial cells from non-obese subjects. The ensemble of dysregulated immune responses in obese subjects during the first three days of infection described here may be a pre-requisite for the cytokine storm development that is more frequently observed in this high-risk group for severe COVID-19. This may shed light on understanding the immune response in obese patients, which may aid in developing better therapeutic strategies to care for obese COVID-19 patients.

## 4. Materials and Methods

### 4.1. Cell Culture

Normal human primary bronchial epithelial (NHBE) cells from non-obese and obese subjects were purchased from a commercial source (MatTek, Ashland, MA, USA) or obtained from the Biobank of the Quebec Respiratory Health Research Network at the Meakins-Christie Laboratories, Research Institute of the McGill University Health Centre (Glen site). Table 1 represents subjects’ characteristics from the non-obese and obese donors. NHBE cells were cultured in BEGM media (Lonza, Walkersville, MD, USA) supplemented with 1% antibiotic antimycotic solution (Wisent, St-Bruno, QC, Canada) in tissue culture flasks coated with Type 1 rat tail collagen (Sigma-Aldrich, Oakville, ON, Canada). Cells were harvested at 90% confluency.

### 4.2. SARS-CoV-2 Infection of NHBE Cells

NHBE cells were seeded in 24-well plates coated with Type 1 rat tail collagen (Sigma-Aldrich, ON, Canada) at a density of 5 × 10^4^ cells per well. Cells were cultured for an average of 4 days in BEGM media (Lonza, Walkersville, MD, USA) supplemented with 1% antibiotic antimycotic solution (Wisent, St-Bruno, QC, Canada). Prior to infection cell quantification was preformed using a Millipore Scepter automated cell counter (MilliporeSigma, Burlington, MA, USA). After which the media on the cells was removed and 200 µL of fresh media was added. A Multiplicity of Infection (MOI) of 0.5 was used to infect the bronchial epithelial cells with SARS-CoV-2/RIM-1 (Genebank accession number: MW599736). A specific amount of virus stock was lysed in RNA extraction reagent for quantification. Cells were harvested at 4-h post-viral incubation or infection media was removed and 200 µL of fresh media was added.

The cells were then incubated for 8, 12, 18, 24, 48 and 72 h. Media were collected at each time point and stored at −80 °C for later analysis. RNA extraction reagent (500 µL) was added to each well and incubated at room temperature for 5 min. Samples were stored at −80 °C until RNA extraction.

### 4.3. RNA Extraction

Total RNA extraction from NHBE cells was performed using a phenol-chloroform extraction (RiboZol RNA extraction reagent, VWR, Leicestershire, UK), as directed by the manufacturer’s instructions. RNA was also extracted from 100 ul of cell culture supernatant, using 1 mL of extraction reagent. Glycogen precipitated the RNA from these samples (ThermoScientific, Waltham, MA, USA), as directed in the manufacturer’s instructions. Contaminating DNA was removed from 500 ng of total RNA using DNAse I and Ribolock RNAse Inhibitor (ThermoScientific, Waltham, MA, USA). This solution was incubated at 37 °C for 30 min. An amount of 1 µL of 50 mM EDTA was added to each sample and incubated at 65 °C for 10 min to inactivate the enzyme. Reverse transcription was performed using AdvanTech 5X Reverse Transcriptase Mastermix (Diamed, Mississauga, ON, Canada).

Viral detection and replication were assessed by measuring levels of SARS-CoV-2 nucleocapsid (N) and viral envelope (UpE) mRNA, respectively. A standard curve generated in the Bio-Rad CFX Maestro 2.2 software of the serially diluted single-stranded complementary DNA (cDNA) extracted from the viral stock was used to quantify the relative levels of these viral genes by determining the corresponding median tissue culture infectious dose (TCID50) of each sample [41]. Viral genes were measured using AdvanTech 2X qPCR MasterMix (Diamed, Mississauga, ON, Canada) in CFX96 thermal cycler (BioRad, Hercules, CA, USA).

### 4.4. RNA Sequencing

RNA from qPCR experiments above was used for total RNA sequence. RNA from NHBE cells from non-obese, obese subjects, non-infected (*n* = 4) and infected (*n* = 4) and obese (*n* = 4) were pooled together for the following time points: 4, 8, 12, 24, 48 and 72 h, for a total of 24 samples. The sequencing was performed by the Centre d’expertise et de services Génome Québec. Their procedure is as follows.

Total RNA was quantified using a NanoDrop Spectrophotometer ND-1000 (NanoDrop Technologies, Inc., Wilmington, DE, USA) and its integrity was assessed on a 2100 Bioanalyzer (Agilent Technologies, Santa Clara, CA, USA). Libraries were generated from 10 ng of total RNA as follows. cDNA synthesis was achieved with the NEBNext RNA First Strand Synthesis and NEBNext Ultra Directional RNA Second Strand Synthesis Modules (New England BioLabs, Ipswich, MA, USA). The remaining library preparation steps were carried out using the NEBNext Ultra II DNA Library Prep Kit for Illumina (New England BioLabs, Ipswich, MA, USA). Adapters and PCR primers were purchased from New England BioLabs. Libraries were quantified using the Kapa Illumina GA with Revised Primers-SYBR Fast Universal kit (Kapa Biosystems, Wilmington, MA, USA). The average size fragment was determined using a LabChip GX (PerkinElmer, Waltham, MA, USA) instrument.

The libraries were normalized and pooled and then denatured in 0.05 N NaOH and neutralized using HT1 buffer. The pool was loaded at 200 pM on an Illumina NovaSeq S4 flowcell as per the manufacturer’s recommendations. The run was performed for 2 × 100 cycles (paired-end mode). A phiX library was used as a control and mixed with libraries at 1% level. Base calling was performed with RTA v3.4.4. Program bcl2fastq2 v2.20 was then used to demultiplex samples and generate FASTQ reads.

### 4.5. Identification of Differentially Expressed Genes (DEGs)

Generated FASTQ files were uploaded to AltAnalyze to directly produce gene-expression estimates using the kallisto program for ultra-fast pseudoalignment and isoform quantification from RNA-Seq FASTQ files [42]. The resultant normalized gene expression (quantile) for each sample was then used to identify differentially expressed genes. As the samples per group were pooled to make the quantity of RNA sufficient to run RN-Seq, each pool was considered as one sample for downstream analysis. Pooling RNA samples was a good option to optimize cost and maintain power. To identify DEGs among different patient groups and different time points of infection, comparing two conditions without replicates (TCWR) has been used by applying individualized DEG (iDEG) method following a distribution calculated across a local partition of related transcripts at baseline expression; thereafter, the probability of each DEG was estimated by empirical Bayes with local false discovery rate control using a two-group mixture model. DEGs were shown if they had less than or equal to 0.05 with fold change more than 2 or less than −2 between the samples examined. The top DEGs between obese and non-obese, infected and non-infected during different time points were listed and the corresponding enriched pathways were listed using Metascape online tool (https://metascape.org/gp/index.html, accessed on 8 February 2023).

### 4.6. Inflammatory Marker Quantification Using qRT-PCR

Real time primers are listed in Table 2. cDNA was synthesized as mentioned above and mRNA levels of inflammatory markers and ACE2 were measured using AdvanTech 2X qPCR MasterMix (Diamed, Mississauga, ON, Canada) in CFX96 thermal cycler (BioRad, Hercules, CA, USA). PCR amplification steps were as follows: 94 °C for 1 min, followed by 40 cycles of 95 °C for 30 s and 58 °C for 1 min. At the end of the amplification, a melting temperature analysis of the amplified gene products was performed routinely for all cases: the PCR products were melted by gradually increasing the temperature from 65 °C to 95 °C in 0.5 °C increments. The ∆∆Cq method was used to measure gene expressions of inflammatory markers and ACE2 after normalizing to the reference genes, GAPDH.

### 4.7. Statistics

One-way ANOVA test and Tukey post hoc test were used to define statistical significance.

## Figures and Tables

**Figure 1 ijms-24-06729-f001:**
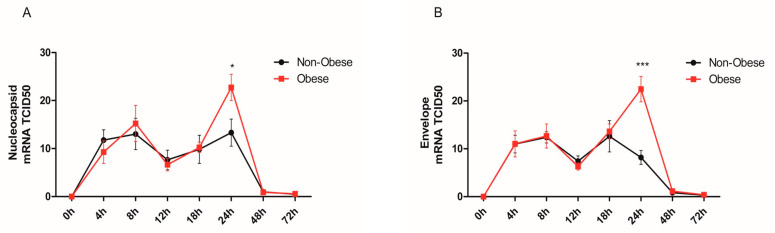
Increased SARS-CoV-2 replication in human bronchial epithelial cells (NHBE) from obese subjects. SARS-CoV-2 infection of bronchial epithelial cells (NHBE) with MOI 0.5. (**A**) Viral load was assessed by measuring nucleocapsid mRNA, *n* = 4/group; (**B**) replication was evaluated by measuring the SARS-CoV-2 envelope (UpE) gene, which was significantly higher in NHBE cells from obese subjects (*n* = 4) compared to non-obese (*n* = 4) subjects. One-way ANOVA test and Tukey post hoc, * *p* < 0.05; *** *p* < 0.001.

**Figure 2 ijms-24-06729-f002:**
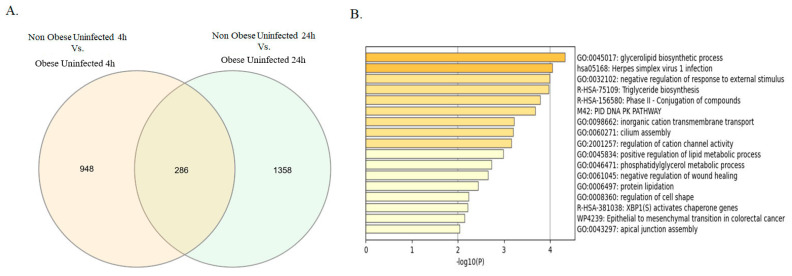
Differentially expressed genes between bronchial epithelial cells from non-infected non-obese and obese subjects. (**A**) Venn diagram showing the intersection of DEGs found between non-infected bronchial epithelial cells from non-obese and obese subjects at 4 and 24 h. (**B**) Enrichment analysis of intersecting genes found in Venn diagram.

**Figure 3 ijms-24-06729-f003:**
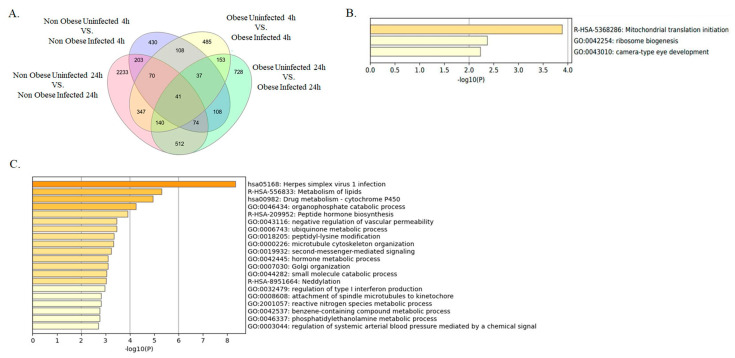
Core transcriptome of bronchial epithelial cells during SAR-CoV-2 infection. (**A**) Venn diagram of intersecting DEGs from non-infected, infected, bronchial epithelial cells from non-obese and obese at 4 and 24 h post-infection. (**B**) Enrichment analysis of the 41 genes found in the intersection of all groups of Venn diagram. This being the core transcriptomic of infected vs. non-infected (non-obese and obese). (**C**) Enrichment analysis of the 728 genes found in the obese uninfected 24 h vs. obese infected 24 h group of the Venn diagram.

**Figure 4 ijms-24-06729-f004:**
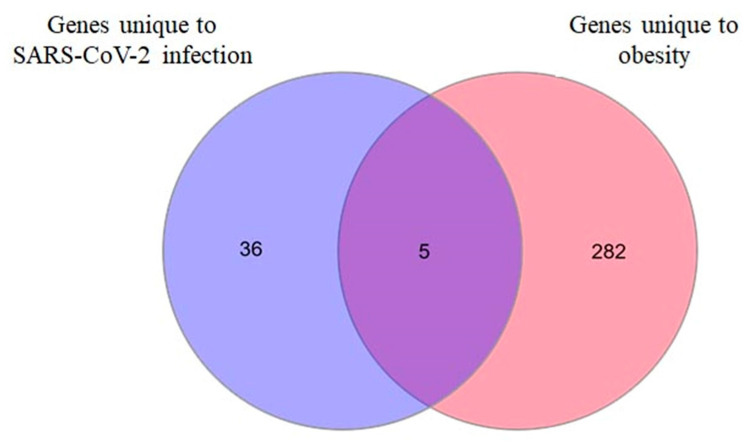
Five DEGs were identified as unique to bronchial epithelial cells from obese subjects and are core players in SARS-CoV-2-infected cells. Venn diagram of intersecting DEGs unique to SARS-CoV-2 infection and associated to obesity. Five genes were identified: *CASC5 (KNL1)*, *LPIN2*, *MIR143HG (CARMN)*, *ULBP1* and *THRB*.

**Figure 5 ijms-24-06729-f005:**
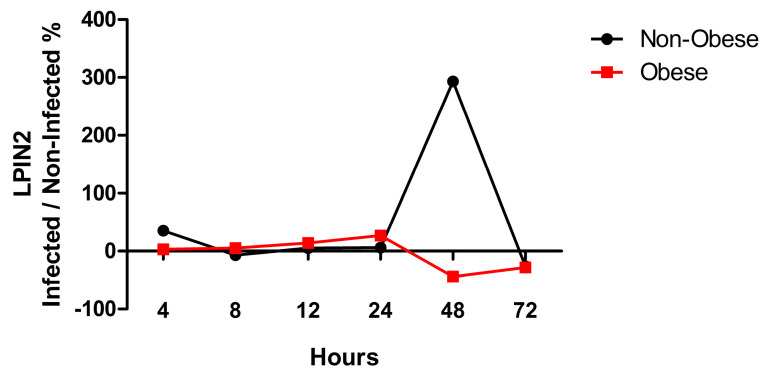
Differential expression of *LPIN2* over infection time between cells from non-obese and obese subjects. The percentage of change of *LPIN2* expression in infected compared to non-infected NHBE cells.

**Figure 6 ijms-24-06729-f006:**
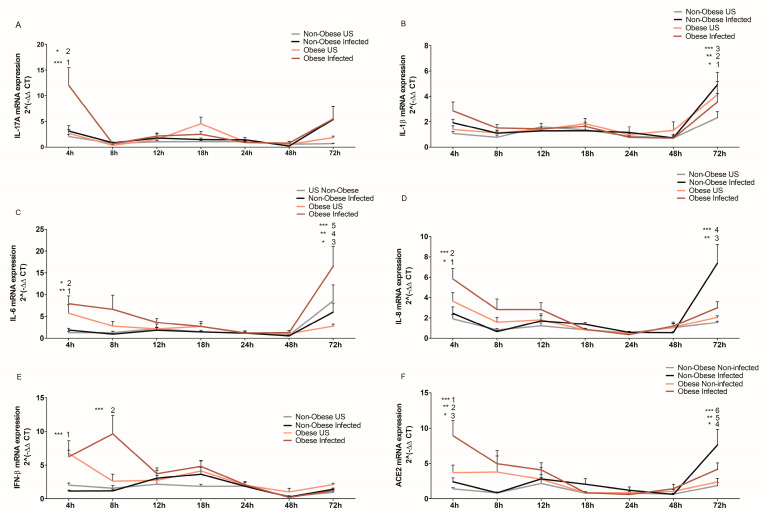
Inflammatory responses in SARS-CoV-2 infection of bronchial epithelial cells. mRNA expression of (**A**) *IL-17A*, (1) obese uninfected vs. obese infected at 4 h, non-obese infected vs. obese infected at 4 h, obese infected at 4 h vs. 8 h, obese infected at 4 h vs. 12 h, obese infected at 4 h vs. 18 h, obese infected at 4 h vs. 24 h, obese infected at 4 h vs. 48 h, (2) obese infected at 4 h vs. 72 h. (**B**) *IL-1β*, (1) non-obese infected at 4 h vs. 72 h, non-obese infected at 18 h vs. 72 h, (2) non-obese infected at 12 h vs. 72 h, non-obese infected at 8 h vs. 72 h, non-obese infected at 24 h vs. 72 h, (3) non-obese infected at 48 h vs. 72 h. (**C**) *IL-6* (1), non-obese infected vs. obese infected at 4 h (2) obese infected at 4 h vs. 48 h, (3) obese infected at 4 h vs. 72 h, (4) obese infected at 8 h vs. 72 h, non-obese infected at 72 h vs. obese infected 72 h, (5) obese infected at 12 h vs. 72 h. (**D**) *IL-8* (1) obese infected at 4 h vs. 12 h, (2) non-obese infected vs. obese infected at 4 h, obese infected at 4 h vs. 18 h, obese infected at 4 h vs. 24, obese infected at 4 h vs. 48, (3) non-obese infected vs. obese infected at 72 h, (4) non-obese uninfected vs. non-obese infected at 72 h, non-obese infected at 4 h vs. 72 h, non-obese infected at 8 h, 12 h, 18 h, 24 h, 48 h vs. 72 h. (**E**) *IFN-β*, (1) non-obese infected vs. obese infected at 4 h, (2) non-obese infected vs. obese infected. (**F**) *ACE2*, (1) obese non-infected vs. obese infected at 4 h, non-obese infected vs. obese infected at 4 h, obese infected at 4 h vs. 48 h, obese infected at 4 h vs. 12 h, obese infected at 4 h vs. 18 h, (2) non-obese infected at 4 h vs. 72 h, (3) obese infected at 4 h vs. obese infected at 12 h, (4) non-obese infected at 18 h vs. non-obese infected 72 h, (5) non-obese infected at 24 h vs. 72 h, non-obese infected at 48 h vs. 72 h, non-obese uninfected at 72 h vs. non-obese infected at 72 h, (6) non-obese infected at 8 h vs. non-obese infected at 72 h. *n* = 4/group. One-way ANOVA test and Tukey post hoc, * *p* ≤ 0,05, ** *p* < 0.01, *** *p* < 0.001.

**Table 1 ijms-24-06729-t001:** Demographic of normal human bronchial epithelial cells from non-obese and obese subjects.

	Non-Obese	Obese
N	4	4
Age, y	42.3 ± 7.5	32 ± 6.6
BMI, kg/m^2^	23.6 ± 5.3	37.1 ± 2.8
Sex (Male/female)	2/2	3/1

Definition of abbreviation: BMI = body mass index. Values shown are mean ± SEM.

**Table 2 ijms-24-06729-t002:** Forward and reverse primers.

*Primer Name*	Oligo Sequence (5′ to 3′)
*SARS-CoV2 N-* *F*	AAGCTGGACTTCCCTATGGTG
*SARS-CoV2 N* *-* *R*	CGATTGCAGCATTGTTAGCAGG
*SARS2-UpE-* *F*	ATTGTTGATGAGCCTGAAG
*SARS2-UpE-* *R*	TTCGTACTCATCAGCTTG
*ACE2-* *F*	TACTGTGACCCCGCATCTCT
*ACE2-* *R*	TCCAACAGTTTCTGTCCAGC
*IL-1β-* *F*	TACATCAGCACCTCTCAAGCA
*IL-1β-* *R*	CCACATTCAGCACAGGACTCT
*GAPDH-* *F*	GAAGGTGAAGGTCGGAGT
*GAPDH-* *R*	GAAGATGGTGATGGGATTTC
*IL-17A-* *F*	GAGGACAAGAACTTCCCCCG
*IL-17A-* *R*	CATTGCCGTGGAGATTCCAAG
*IFN-β-* *F*	CTTGGATTCCTACAAAGAAGCAGC
*IFN-β-* *R*	TCCTCCTTCTGGAACTGCTGCA
*IL-8-* *F*	TCTGCAGCTCTGTGTGAAGGTG
*IL-8-* *R*	AATTTCTGTGTTGGCGCAGTG
*IL-6-* *F*	ACCTTCCAAAGATGGCTGAAA
*IL-6-* *R*	GCTCTGGCTTGTTCCCTCACTAC

## Data Availability

All data are contained in manuscript.

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
