# Peer review of "Lung Epithelial Cells from Obese Patients Have Impaired Control of SARS-CoV-2 Infection"

_ijms, 2023, doi:10.3390/ijms24076729_

Round 1
Reviewer 1 Report
ManuscriptID: ijms-2242280
Title: Lung epithelial cells from obese patients have impaired control of SARS-CoV-2 infection
Comment for the authors
The article entitled "Lung epithelial cells of obese patients have impaired control of SARS-CoV-2 infection" aims to investigate the mechanisms that make obesity a risk factor of severe COVID-19 using an in vitro model and RNA sequencing. The results show that normal human bronchial epithelial cells (NHBE) from obese subjects infected with SARS-CoV-2 show increased viral load and replication compared to those from non-obese subjects. Using transcriptomic profiling via RNA sequencing, LPIN2, an inflammasome regulator, was found to be a unique differentially expressed gene in infected NHBE cells from obese subjects. Furthermore, the expression of five inflammatory markers (IL-17A, IL-1BETA, IL-6, IL-8, IFN-BETA) and ACE2 is higher in infected NHBE cells from obese than non-obese subjects.
The manuscript is well organized and written in a clear and concise manner. The purpose of the study is clearly stated. The results are interesting and the conclusions are supported by experimental data. These data may be useful in elucidating the association between obesity and severe COVID-19.
General comment
Title: Since bronchial cells were used in the study, it may be more appropriate to use the term "bronchial epithelial cells" rather than “pulmonary epithelial cells”.
Throughout the text, NHBE cells have been defined as both lung epithelial cells and bronchial epithelial cells. It would be more correct to define them in one way and that is as bronchial epithelial cells.
Generally, the figures and the legends are organized effectively, with the exception of figure 6 which is difficult to read due to the too small size of both the numbers and the characters of the legends.
Furthermore, the references in the text relating to figure 6 are all incorrect and therefore must be rewritten. For example: the exact reference for IFN-beta is figure 6F and not figure 6A as indicated in the text. The same goes for all other markers.
In addition, in the text, ACE2 expression is referred to as figure 7, which does not exist, instead of figure 6F.
Author Response
We would like to thank the reviewer for their valuable comments and suggestions. We have addressed all the comments.
1) We have ensured that "bronchial epithelial cells" is consistent through out the manuscript.
2) Modifications had been made to figure 6 to improve visibility of the panel. We have also corrected the figures to correspond to the figure legend.
3) Thank you for pointing this out. The legend has been revised.
Reviewer 2 Report
The authors reported their experience on the association between the mechanisms of SARS-CoV-2 infection and obesity providing an intriguing molecular overview.
The authors used an in vitro live infection model and RNA-sequencing in order to understand the effect of obesity on the course of SARS-CoV-2 infection. This study revealed the enhancement of viral load and replication in bronchial epithelial cells (NHBE) from obese subjects at 24h of infection (MOI=0.5) as compared to non-obese. Transcriptomic profiling via RNA-seq highlighted the enrichment of lipid metabolism related-pathways along with LPIN2, an inflammasome regulator, as a unique differentially expressed gene (DEG) in infected bronchial epithelial cells from obese subjects. Such
findings correlated with altered cytokine and angiotensin converting enzyme-2 (ACE2) expression during infection of bronchial cells.
The paper is well written and easy to follow.
The topic is really fascinating.
I have no minor nor major concerns with regard to this manuscirpt.
Author Response
We would like to thank the reviewer for their comments.